# Whispered Speech Conversion Based on the Inversion of Mel Frequency Cepstral Coefficient Features

**Qiang Zhu** [1], **Zhong Wang** [1,*], **Yunfeng Dou** [2] and **Jian Zhou** [2]

1   School of Computer Science and Technology, Hefei Normal University, Hefei 230601, China; qiangzhu@hfnu.edu.cn
2   School of Computer Science and Technology, Anhui University, Hefei 230601, China; wangz@hfnu.edu.cn (Y.D.); jzhou@ahu.edu.cn (J.Z.)
*   Correspondence: zhongw@ustc.edu.cn; Tel.: +86-189-0560-7916

**Abstract:** A conversion method based on the inversion of Mel frequency cepstral coefficient (MFCC) features was proposed to convert whispered speech into normal speech. First, the MFCC features of whispered speech and normal speech were extracted and a matching relation between the MFCC feature parameters of whispered speech and normal speech was developed through the Gaussian mixture model (GMM). Then, the MFCC feature parameters of normal speech corresponding to whispered speech were obtained based on the GMM and, finally, whispered speech was converted into normal speech through the inversion of MFCC features. The experimental results showed that the cepstral distortion (CD) of the normal speech converted by the proposed method was 21% less than that of the normal speech converted by the linear predictive coefficient (LPC) features, the mean opinion score (MOS) was 3.56, and a satisfactory outcome in both intelligibility and sound quality was achieved.

**Keywords:** whispered speech conversion; MFCC feature inversion; Gaussian mixture model; cepstral distortion

## 1. Introduction

Whispered speech is a method of articulation different from normal speech [1]; it is produced without vibration of the vocal cords at a low sound level, which causes the voiced sound of whispered speech to have no fundamental frequency and an energy 20 dB less than that of normal speech [2]. Because of these characteristics, whispered speech is widely used in places where loud noises are prohibited such as conference rooms, libraries, and concert halls. Because of its potential applications [3–6], in recent years, studies on whispered speech (e.g., whispered speech emotion recognition, whispered speech enhancement, whispered speech recognition, and whispered speech conversion) have gradually attracted researchers' attention. Among them, whispered speech conversion, a technique that converts whispered speech to normal speech, has been widely used in mobile communication, medical equipment, security monitoring, and crime identification [7–9]. For example, for a laryngeal cancer patient who has undergone a laryngectomy, the conversion of whispered speech to normal speech can greatly improve the patient's speech communication experience and efficiency [10].

At present, whisper speech conversion is mainly divided into two categories. One is rule-based whisper conversion, mainly using empirical observation or statistical modeling to generate transformation rules. The other one is machine learning based, which includes Gaussian mixture model and neural network method [11]. In terms of the conversion of whispered speech to normal speech, researchers used multiple excited linear prediction method to implement Chinese speech reconstruction [12,13], in which normal speech is reconstructed by adding fundamental frequencies to the speech energy based on a whispered speech formant. However, the formant distribution of whispered speech is not

identical to that of normal speech, and its formant frequencies are shifted to high frequencies [14]; therefore, the sound quality of the reconstructed speech was poor. Perrotin et al. adopted the mixed excitation linear prediction (MELP) model [15] to reconstruct whispered speech, where the whispered speech is divided into five frequency bands. Of these five frequency bands, four low-frequency bands were treated as voiced excitation, and the high-frequency band was treated as unvoiced excitation, to which fundamental frequencies are added based on the energy of speech, modifying the characteristics of the formant spectrum. The advantage of this method is its easy application to communication systems; however, because the added fundamental frequency is more monotonous than normal speech, the reconstructed speech often sounds "metallic". Sharifzadeh et al. improved the code-excited linear prediction (CELP) method by adding a whispered speech preprocessing module, a fundamental frequency-generation module and a formant modification module to achieve the conversion of whispered speech to normal speech. Their model has very limited effectiveness on continuous speech and is prone to generate quantization errors when performing vector quantization on an excitation source [16]. Xu et al. proposed a whispered speech reconstruction system based on homomorphic signal processing and relative entropy segmentation [17,18], which can better eliminate the metallic sound caused by adding fundamental frequencies, regardless of rhyme.

The above methods are all rule-based whispered speech conversion techniques. In recent years, an increasing number of scholars have begun to use the statistical distribution characteristics of speech features to convert speeches and adopt probabilistic methods to achieve the prediction of target feature vectors from source feature vectors. Among them, the Gaussian mixture model (GMM) is the most widely used. Toda et al. applied the GMM-based speech conversion methods to whispered speech conversion for the first time [19–23], making it possible to convert whispered speech to normal speech. In addition, Chen et al. proposed to build a continuous probability model for the spectral envelope of whispered speech using a probability-weighted GMM [24] to obtain the mapping relations of channel parameters between the whispered speech and the corresponding normal speech. The probability-weighted GMM can theoretically obtain a reasonable mapping relation between the source and target feature vectors, but it is still difficult for GMM to build a model because of the many dimensions of the spectral envelope. Therefore, it is important to choose speech features when converting whispered speech using GMM. Compared with other feature parameters, the Mel frequency cepstral coefficient (MFCC) simulates the hearing characteristics of the human ear. Boucheron et al. used MFCC feature inversion to synthesize speech, and the perceptual evaluation of speech quality (PESQ) of the synthesized speech was over 3.5 [25]. To consider the sparseness of speech, in this study, we used the L1/2 algorithm to reconstruct speech based on the research by Boucheron et al., which improved the reconstruction effectiveness.

In this paper, we report a method for converting whispered speech to normal speech based on MFCC and GMM. In the model establishment stage, we extracted the MFCC parameters of each frame of whispered speech and reference normal speech from the parallel corpus; we then used the GMM to establish the joint probability distribution between the frame feature parameters. In the conversion stage, we first input the frame feature parameters of whispered speech into the model. After estimating the feature parameters of normal speech, we used the MFCC feature parameter inversion method to directly reconstruct normal speech. Compared with existing GMM-based whispered speech conversion methods, the GMM developed in this study took into account the correlation between adjacent frames of speech, and the proposed method does not require fundamental frequency estimation when reconstructing speech.

## 2. Whispered Speech Conversion Process

### 2.1. GMM-Based Whispered Speech Conversion Model

GMM-based whispered speech conversion reconstructs normal speech from whispered speech by estimating the acoustic parameters of normal speech. As shown in Figure 1, the method has two stages: the model establishment stage and the conversion stage. In the first stage, the MFCC parameters of each frame of normal speech and whispered speech were extracted, and then the reference normal speech and whispered speech were processed with dynamic time warping (DTW). The normal speech-whispered speech joint MFCC feature distribution model was then constructed using GMM. In the second stage, the MFCC features of whispered speech were converted to the MFCC features of normal speech using the established GMM from the first stage, and the normal speech was then synthesized using the MFCC features of normal speech.

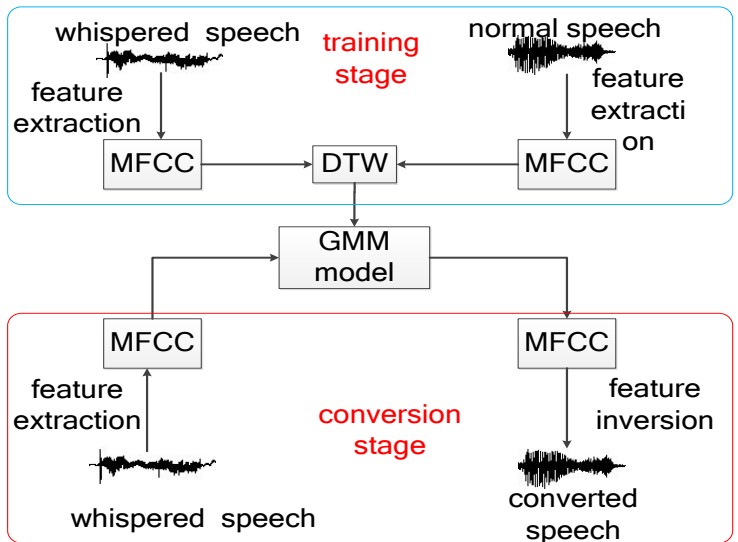

**Figure 1.** Flow chart of whispered speech–normal speech conversion.

The probability density function of an M-order GMM was obtained by a weighted summing of M Gaussian probability density functions as follows:

$$P(X/\lambda) = \sum_{i=1}^{M} w_i b_i(X) \tag{1}$$

where $X$ is a D-dimensional random vector; $b_i(X_t)$ is the subdistribution ($i = 1, \ldots, $ M); $w_i$ is the mixed weight ($i = 1, ..., $ M). Each subdistribution is a D-dimensional joint Gaussian probability distribution, which can be expressed as follows:

$$b_i(X) = \frac{1}{(2\pi)^{D/2}|\Sigma_i|^{1/2}} \exp\left\{-\frac{1}{2}(X - \mu_i)^t \Sigma_i^{-1}(X - \mu_i)\right\} \tag{2}$$

where $\mu_i$ is the mean vector, and $\Sigma_i$ is the covariance matrix.

Assuming that the source feature vector and the target feature vector conform to the joint Gaussian probability distribution, the GMM was used to model the mixed MFCC feature parameters as follows:

$$p\left(\begin{array}{c} x \\ y \end{array}\right) = \sum_{m=1}^{M} a_m N\left(\left(\begin{array}{c} x \\ y \end{array}\right), \mu_m, \Sigma_m\right) \tag{3}$$

$$\mu_m = \left(\begin{array}{c} \mu_m^x \\ \mu_m^y \end{array}\right), \quad \Sigma_m = \left(\begin{array}{cc} \Sigma_m^{xx} & \Sigma_m^{xy} \\ \Sigma_m^{yx} & \Sigma_m^{yy} \end{array}\right) \tag{4}$$

where $N\left(\begin{pmatrix} x \\ y \end{pmatrix}, \mu_m, \Sigma_m\right)$ is the $m$th Gaussian component, where $\mu_m$ and $\Sigma_m$ are the mean and the covariance, respectively, of the $m$th Gaussian component of the GMM; $\mu_m^x, \mu_m^y$ are the source feature vector corresponding to the $m$th Gaussian component and the mean vector of the target feature vector, respectively; $\Sigma_m^{xx}, \Sigma_m^{yy}$ are the autocorrelation matrixes of the source target feature vector and the target feature vector, respectively, corresponding to the $m$th Gaussian component; $\Sigma_m^{xy}, \Sigma_m^{yx}$ are the covariance matrixes of the source target feature vector and the target feature vector, respectively, corresponding to the $m$th Gaussian component; $a_m$ is the probability weight of the $m$th Gaussian component.

The expectation-maximization (EM) algorithm was used to estimate the parameters of the Gaussian probability distribution. In each Gaussian component, a linear relation between the source feature vector and the target feature vector was developed as follows:

$$\hat{y}_t = \sum_{m=1}^{M} p(c_m | x_t)(A_m x_t + B_m) \tag{5}$$

where $p(c_m | x_t)$ is the posterior probability of the $t$th frame feature vector in the $m$th Gaussian component; $A_m$ is the conversion matrix of the $m$th Gaussian distribution; $B_m$ is the offset vector. Solving $A_m$ and $B_m$ using the least square error criterion yields the following:

$$A_m = \Sigma_m^{yx} \Sigma_m^{xx-1}, B_m = \mu_m^y - A_M^* \mu_m^x \tag{6}$$

$$p(c_m | x_t) = \frac{a_m N(x_t, \mu_m^x, \Sigma_m^{xx})}{\sum\limits_{m=1}^{M} a_m N(x_t, \mu_m^x, \Sigma_m^{xx})} \tag{7}$$

where $a_m$ is the probability weight of the $m$th Gaussian component, and $x_t$ is the source feature parameter vector of the $t$th frame.

### 2.2. Inversion of Speech Features

The MFCC analysis is based on the mechanism of human hearing characteristics, i.e., the analysis of the speech spectrum based on the results of human hearing experiments. For this process, it is a common practice to pass the speech signal through a series of filter banks, which are called Mel filter banks. When the speech signal passes through the filter banks, the output is the power spectrum at the Mel frequency. By performing a discrete cosine transform (DCT) on all the log operation of filter bank outputs, we obtain the MFCC as follows:

$$mfcc = DCT\{\log(z)\} \tag{8}$$

where $Z$ is the energy spectrum of the Mel filter as follows:

$$z = \Phi y \tag{9}$$

The weight matrix $\Phi \in R^{K \times (N/2+1)}$ of the Mel frequency scale is the tone frequency based on the perception by the human ear. The value of the Mel frequency scale exhibits a roughly logarithmic distribution corresponding to the actual frequency. The speech frequency can be divided into a series of triangular filter sequences, i.e., the Mel filter banks, in which $K$ is the spectral line of the Mel filter banks, $y$ is the power spectrum of speech sound $x$ and $X$ is the frequency domain data of speech by STFT, as follows:

$$y = |X|^2 = |F\{x\}|^2 \tag{10}$$

To obtain the reconstructed speech by the inverting MFCC, the process of obtaining MFCC features was inverted. Since the DCT, log, and squaring operations are all reversible, when Z and $\Phi$ are known, it is possible to more accurately estimate the energy spectrum $y$ of the speech from the MFCC features and its phase spectrum [26,27] to obtain the inverted speech frame.

1. Estimation of energy spectrum Y

Because of the sparse characteristics of speech signals, the use of sparse characteristics in their decomposition can effectively avoid interference from some noises, which is of great significance for signal reconstruction. Xu et al. [28] conducted an in-depth study on the sparse decomposition of signals and demonstrated the advantages of the L1/2 norm in the sparseness of signal decomposition, which has achieved good results when applied to the reconstruction of sparse signals. In this study, we propose an L1/2 + 2 double sparse constraint-based signal reconstruction model and a solution for $y$ with known Z and $\Phi$. The objective function model is as follows:

$$\min_{y}\{||\Phi y - z||_2^2 + \sum_i^N (\lambda_2 ||y_i||_{1/2}^{1/2} + \lambda_1 ||y_i||_2^2)\} \tag{11}$$

where $\lambda_1$ and $\lambda_2$ are the regularization parameters that control the sparseness and smoothness, respectively, of the coefficient vector, satisfying $\lambda_1 > 0$ and $\lambda_2 > 0$. From Equation (4), we obtain the updated formula for y as follows:

$$y = (I + \tfrac{\mu\lambda_2}{2}||\cdot||_{1/2}^{1/2})^{-1}((1 - \lambda_1\mu)y + \mu\Phi^T(z - \Phi y))$$
$$\Rightarrow y = H_{\lambda_2,\mu,1/2}(B_\mu(y)) \tag{12}$$

where $\nabla(||\cdot||_{1/2}^{1/2})$ is the gradient of $||y||_{1/2}^{1/2}$; $\Phi^T$ is the transpose of $\Phi$; $(I + \tfrac{\mu\lambda_2}{2}||\cdot||_{1/2}^{1/2})^{-1}$ is the operator of y; $H_{\lambda_2,\mu,1/2}(\cdot) = (I + \tfrac{\lambda_2\mu}{2}\nabla(||\cdot||_{1/2}^{1/2}))^{-1}$; $B_\mu(y) = (1 - \lambda_1\mu)y + \mu\Phi^T(z - \Phi y)$.

According to the literature [29], we set the following:

$$y = \begin{cases} \tfrac{4}{3}[B(y)]_i * (\cos(\tfrac{\pi}{3} - \tfrac{\phi_{\lambda_2,\mu}(y_i)}{3}))^2, & |[B(y)]_i| > \tfrac{\sqrt[3]{54}}{4}(\lambda_2\mu)^{\frac{2}{3}} \\ 0, & \text{otherwise} \end{cases} \tag{13}$$

where $\phi_{\lambda_2,\mu}(y_i) = \arccos\left(\tfrac{\lambda_2\mu}{8}(\tfrac{|y_i|}{3})^{-\frac{3}{2}}\right)$, and $i$ is the number of iterations ($i = 1, 2, 3 \cdots$). By the method, the energy spectrum can be directly calculated from the MFCC spectrum without fundamental frequency estimation.

2. Phase spectrum estimation

We used the least squares method to estimate the speech frame from the power spectrum, i.e., the power spectrum was square-rooted and then converted into an amplitude spectrum. An appreciable amount of phase information is lost when obtaining the power spectrum by feature extraction; thus, it is important to recover the phase spectrum. The least square method with the inverse short-time Fourier transform magnitude (LSE-ISTFTM) algorithm was used to modify the discrete Fourier transform (DFT) and inverse discrete Fourier transform (IDFT) to estimate the phase spectrum [30], which, in combination with the given amplitude spectrum, was then used to estimate the speech for each frame in the time domain. Finally, the speech was reconstructed by adding and overlapping the sequence of speech frames. The LSE-ISTFTM Algorithm 1 process is as follows:

---

**Algorithm 1** LSE-ISTFTM

---

1:    Input: Amplitude spectrum, $\left|\widetilde{X}\right|$; Number of iterations, M
2:    Output: Estimated speech frame, $\hat{x}$
3:    Initialize $\hat{x}^{(1)}$ to white noise
4:    While $k \leq M$, do
5:    $\hat{X}^{(k)} \leftarrow F\{\hat{x}^{(k)}\}$
6:    $\hat{X}^{(k)} \leftarrow \left|\widetilde{X}\right| e^{j\angle \hat{X}^{(k)}}$
7:    $\hat{x}^{(k+1)} \leftarrow F^{-1}\{\hat{X}^{(k)}\}$
8:    $k \leftarrow k+1$
9:    End while

---

## 3. Simulation Experiment

### 3.1. Experimental Conditions

The experiment was performed on a PC with a Core i5 3.2 GHz processor and 4 G memory, using the MATLAB 2013a simulation software. To verify the effectiveness of the proposed algorithm, we used 100 sentences that had been recorded in a quiet environment. There were 10 speakers (five females and five males), each with 10 sentences, covering five subjects such as stock market, catering, tourism, sports and film. The length of each sentence was approximately 2 s, with a sampling rate of 8 kHz and a precision of 16 bit. The frame segmentation was performed on the signals, with a frame length of 512 sampling points and an interframe overlap of 50%, three frames of whispered speech and one frame of normal speech constituted a joint feature vector. We randomly selected 90% whispered speech sentences and the corresponding parallel corpus as training data, the rest (10%) was used as test data. The model does not consider speaker relevance. Because the speech information of the current frame of speech has a close relationship with the frame before and after the current frame, to achieve better conversion effectiveness, we not only used the current speech frame after DTW but also considered the frames before and after the speech frame. Also, under the condition that the pronunciation is standard Chinese, if the training set and test set are different subjects, even unseen speaker, it has little impact on the final result.

To verify the effectiveness of the proposed L1/2 algorithm, we compared it with the L2 algorithm; the results are shown in Figure 2, which the number of EM iterations is 300.

The data in Figure 2 show that the effectiveness of the proposed algorithm on speech reconstruction has been greatly improved compared to that of the L2 algorithm. We generated the spectrograms of the two algorithms after the inversion to further validate the advantages of the proposed method, as shown in Figure 3.

The data in Figure 3 show that compared to the L2 algorithm, the reconstructed speech using the proposed algorithm was clearer, with reduced interference from noise. The proposed method obtained the spectrograms similar to the reference normal speech without estimating the fundamental frequency. The harmonic component can be clearly seen from the spectrograms, indicating that the proposed method can obtain the spectrograms estimation of normal speech. We thus performed subsequent experiments using the proposed algorithm.

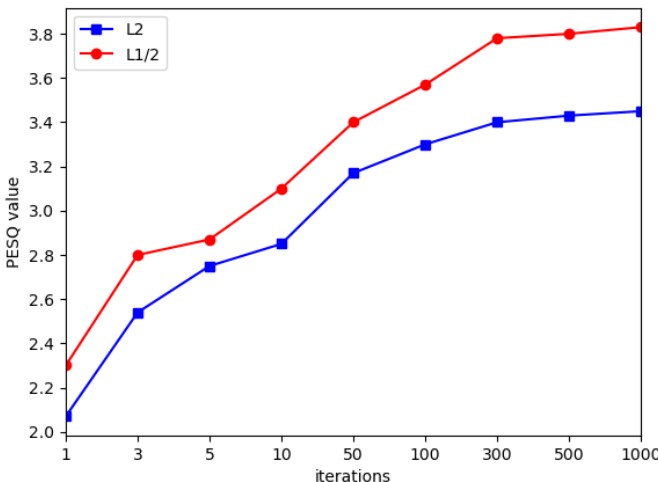

**Figure 2.** Experimental comparison of the L2 and L1/2 methods with different iteration numbers in speech inversion.

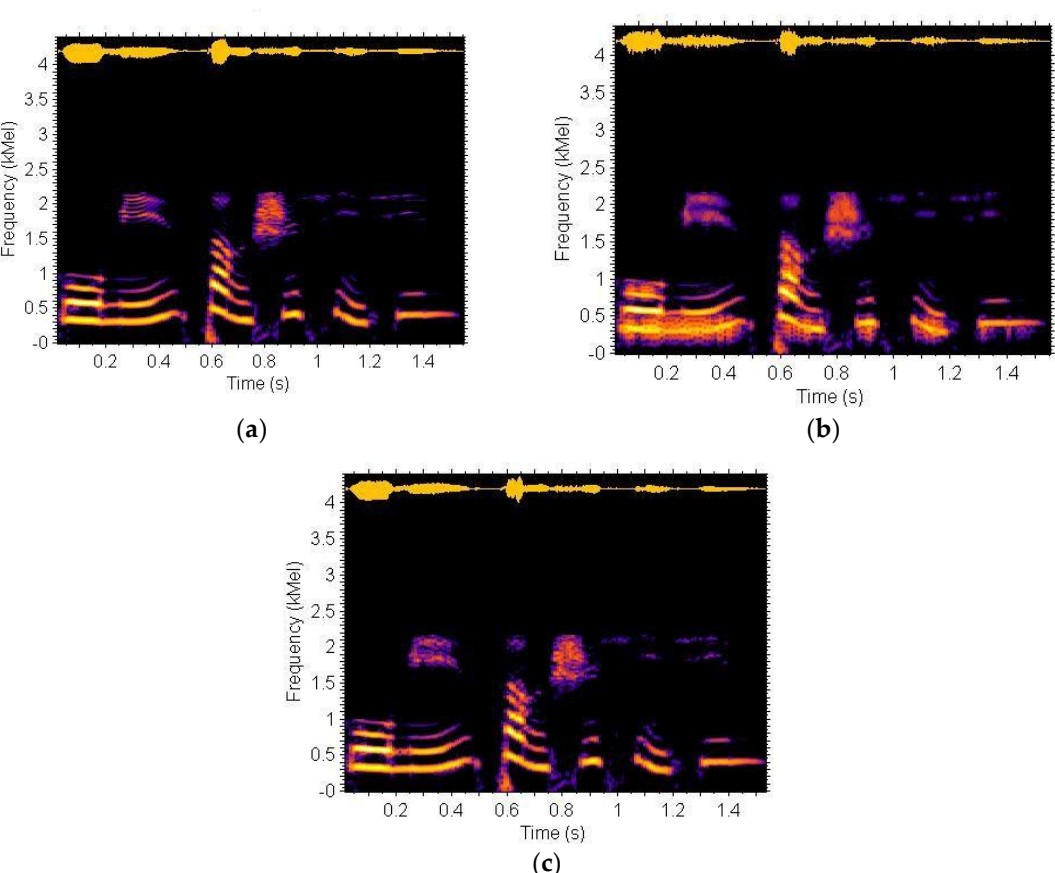

**Figure 3.** Comparison of the spectrograms of two algorithm: (**a**) original speech; (**b**) L2 algorithm; (**c**) proposed algorithm.

In the next experiment, we examined the performance of the proposed method both subjectively and objectively. The subjective test mainly used the auditory characteristics of the human ear to evaluate the converted speech from multiple aspects, such as propensity, intelligibility, and naturalness, using the mean opinion score (MOS) scale. The MOS scale categorizes speech quality into five levels, i.e., 1: Bad, unbearable annoying; 2: Poor, obviously distorted and annoying; 3: Fair, perceivably distorted and slightly annoying; 4: Good, slightly distorted; 5: Excellent, no perceivable distortion. The subjects were asked to score the converted speech based on the above scale to assess its quality. For the objective test, we used the cepstral distortion (CD), which is calculated by the following formula:

$$CD = \frac{10}{\log 10} \sqrt{2 \sum_{d=1}^{D} (C_d - C_d')^2} \qquad (14)$$

where $C_d$ and $C_d'$ are the $d$th dimensional Mel frequency cepstra of the converted speech and the target speech, respectively, and $d$ is the number of speech frames. The average value of the frames is used as the CD value of the speech segment.

The number of Gaussian components of GMM (M) must be determined first, and it is difficult to deduce the optimal M theoretically; however, that number can be experimentally obtained based on different data sets. It is generally set to 4, 8, 16, etc. In this study, from the results in Figure 4, we found the optimal conditions are as follows: multiframe speech and M = 32. Thus, in subsequent experiments, these conditions were adopted.

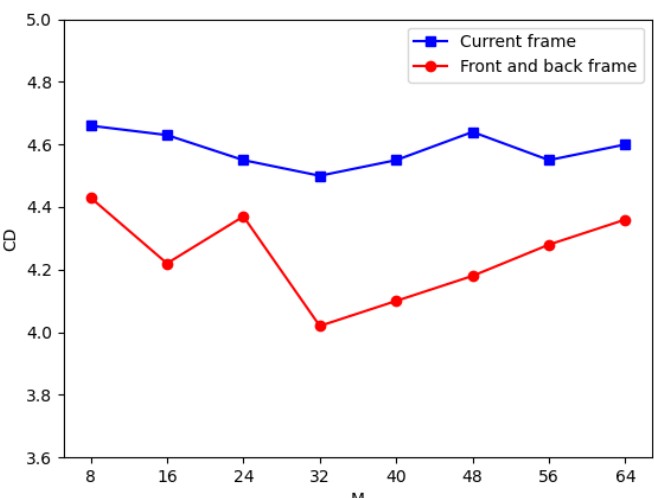

**Figure 4.** Parameter selection curve.

*3.2. Experimental Results*

When using the MOS score for a subjective evaluation, the higher the MOS score, the higher the intelligibility and naturalness of the converted speech. In the MOS evaluation, we invited five testers to score 10 converted speeches from each conversion direction. The results are shown Table 1 in which LPC [31] was used to extract partial correlation coefficients and build signal generation models for synthesis, the network configurations of DNN-based [11] were 60-120-60-120-60, MFCC represents the result from using only the current frame for feature conversion, and MFCCs represents the result from using both the current frame and the previous and subsequent frames (i.e., multiple frames) for feature conversion.

**Table 1.** Comparison of the MOS scores for four methods.

| Conversion Method | Linear Predictive Coefficient (LPC) | DNN | MFCC | MFCCs |
|---|---|---|---|---|
| MOS mean | 3.02 | 3.22 | 3.45 | 3.56 |

The results in Table 1 show that using multiple frames is more effective than using a single frame. At the same time, the MOS scores of the converted whispered speech using the MFCC feature inversion are all greater than 3, indicating the conversion is very effective in recognizing whispered speech.

CD is an objective evaluation standard, especially for evaluating the performance of cepstral feature conversion. According to the formula, the smaller the distance (d), the closer the converted speech to normal speech. As shown in Figure 5, the speech reconstructed using the proposed method was more similar to normal speech, indicating that the feature inversion algorithm is a feasible method for whispered speech conversion.

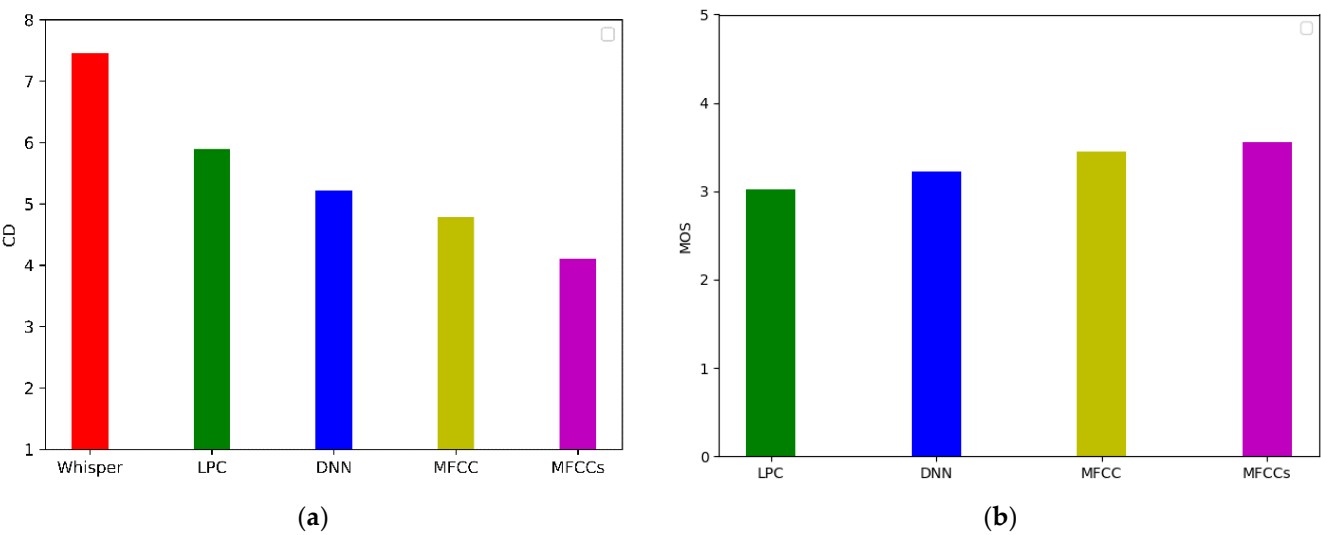

(**a**)                                    (**b**)

**Figure 5.** (**a**) Cepstral distances between normal speech with whispered speech and converted speeches using LPC, DNN, MFCC and MFCCs. (**b**) MOS with 95% confidence w.r.t LPC, DNN, MFCC and MFCCs.

To more visually show the conversion effectiveness, a spectrogram case is shown in Figure 6, in which the speech sentence is "gu min sang shi xin xin" ("shareholders have lost confidence"). The data in Figure 6 show that the MFCC feature inversion-based method proposed in this study achieved a better conversion between formant and spectral envelope. The whispered speech is a voiced sound with an energy approximately 20 dB lower than that of normal speech; however, the sound energy is increased significantly in the converted speech. The speech converted using the proposed method also exhibits a clearer harmonic phenomenon than whispered speech, with significant horizontal bars of the harmonics and distinct alternation of voiced and unvoiced sounds, indicating that the spectrogram of the converted speech is more similar to that of normal speech.

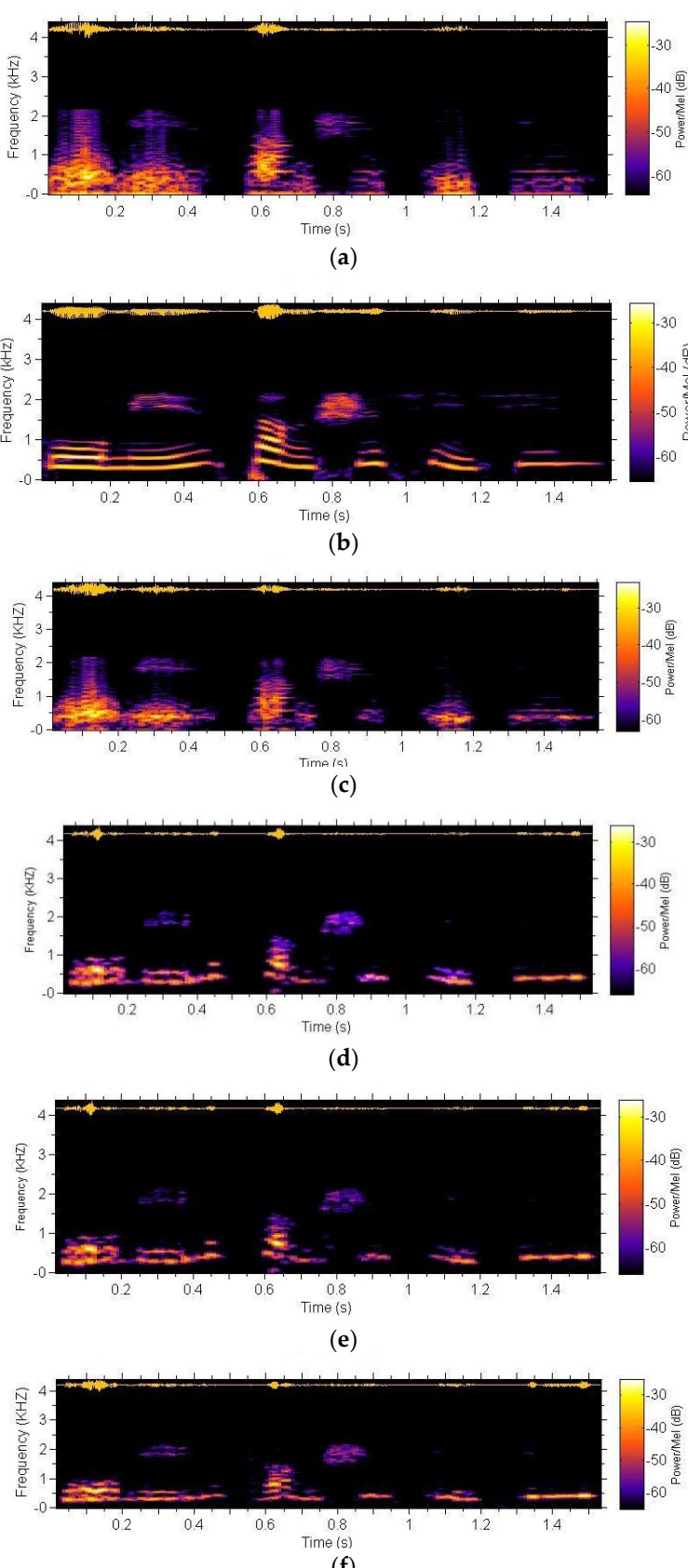

**Figure 6.** Spectrogram of the sentence "gu min sang shi xin" based on MFCC feature conversion: (**a**) whispered speech; (**b**) normal speech; (**c**) speech converted using LPC; (**d**) speech converted using DNN; (**e**) speech converted using MFCC; (**f**) speech converted using MFCCs.

## 4. Conclusions

To consider the sparseness of speech, we proposed to use the L1/2 algorithm to invert the MFCC features, which generates a good hearing effect. We used the GMM to jointly model the feature parameters of whispered speech and normal speech, we obtained a conversion model that converted whispered speech features to normal speech features, and lastly, we adopted feature inversion to reconstruct the converted speech. Experiments showed that the proposed method was very effective in converting whispered speech, with significant improvement in sound quality

**Author Contributions:** Writing—original draft preparation, Q.Z.; writing—review and editing, Z.W.; software, Y.D. and J.Z. All authors have read and agreed to the published version of the manuscript.

**Funding:** This research was funded by the National Natural Science Foundation of China (No. 61976198), the Key Research and Development Plan of Anhui Province (No. 201904d07020012), the Natural Science Research Key Project for Colleges and University of Anhui Province (No. KJ2019A0726), and the High-Level Scientific Research Foundation for the Introduction of Talent of Hefei Normal University (No. 2020RCJJ44).

**Institutional Review Board Statement:** Not applicable.

**Informed Consent Statement:** Not applicable.

**Data Availability Statement:** Not applicable.

**Conflicts of Interest:** The authors declare no conflict of interest.

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
