# Peer review of "Whispered Speech Conversion Based on the Inversion of Mel Frequency Cepstral Coefficient Features"

_algorithms, doi:10.3390/a15020068_

Round 1

Reviewer 1 Report

The introduction does not give a proper overview of the work in the field.

  • citations 3-6 introduced as potential applications of the whispered speech refer to only one such application - speaker detection from whispered speech. Others are about whisper detection or analysis, [6] is not about whisper at all.
  • Authors name many areas, where " converts whispered speech to normal speech, has been widely used" but did not cite any relevant work
  • The same about their claim on lines 36-38
  • lines 39 - 40 - citation of  Liu et al.  - there is not such author int he cited articles.
  • line 62 "The above methods are all rule-based whispered speech conversion techniques" but the citations use attention models and LSTM networks. So what is "rule based" according to authors?
  • line 68 - citation 14-18 - are supposed to be about " GMM-based speech conversion methods to whispered speech conversion", but 15 and 16 are about special architecture of GMM (subspace GMM) and 18 is about speech enhancement.
  • Following citation is stated (liens 69 - 71)  : "Iwama et al. proposed to build a continuous probability model for the spectral envelope of whispered speech using a probability-weighted GMM [19] to obtain the mapping relations of channel parameters between the whispered speech and the corresponding normal speech"  The work does not not contain word whisper at all.

lines 100 - 102  "In the first stage, the reference normal speech and whispered speech were processed with dynamic time warping (DTW), and then the MFCC parameters of each frame of normal speech and whispered speech were extracted." what is used for DTW alignment, when MFCCs are extracted only after it?

line 128 - upper indices needs to be mixed (xy, yx)

line 151  - log operation is missing in the description MFCC computation

line 158-159 - "The speech frequency can be divided into a series of triangular filter sequences..." does not make any sense.

line 162: X is not explained

line 169: "Because of the sparse characteristics of speech signals" -- what is meant by the "sparse characteristics"?

line 171: "Xu et al. [23]" -- there is no author Xu in article 23.

Questions to experimental setup:

  • How many speakers were in the speech database?
  • Did authors use different speakers in train and test set?

How the pairs C, C' for Eq. 14 were created?

The comparison with other methods is missing. It only compares different variant of the same technique.

The proposed method novelty is only to use 3 consecutive frames for the processing, other parts of the method have already be published.

Due to the problems in citations I suggest to reject the paper.

Author Response

Response to Reviewer 1 Comments

Point 1: The introduction does not give a proper overview of the work in the field.

Response 1: Thank you for your suggestion and We add an overview of whisper speech conversion in the introduction section.

Point 2: citations 3-6 introduced as potential applications of the whispered speech refer to only one such application - speaker detection from whispered speech. Others are about whisper detection or analysis, [6] is not about whisper at all.

Response 2: Thank you for pointing out and reference [6] has been replaced as follows:

  1. Galić J, Popović B, Pavlović D Š. Whispered speech recognition using hidden markov models and support vector machines[J]. Acta Polytechnica Hungarica, 2018, 15(5):11-29

Point 3: Authors name many areas, where " converts whispered speech to normal speech, has been widely used" but did not cite any relevant work.

Response 3: We agree with this suggestion and add 3 references [26-28] as follows:

  1. Deng J, Xu X, Zhang Z, et al. Exploitation of phase-based features for whispered speech emotion recognition[J]. IEEE Access, 2016, 4: 4299-4309.
  2. Sardar V M, Shirbahadurkar S D. Timbre features for speaker identification of whispering speech: selection of optimal audio descriptors[J]. International Journal of Computers and Applications, 2021, 43(10): 1047-1053.
  3. Houle N, Levi S V. Acoustic differences between voiced and whispered speech in gender diverse speakers[J]. The Journal of the Acoustical Society of America, 2020, 148(6): 4002-4013.

Point 4: The same about their claim on lines 36-38

Response 4: We agree with this suggestion and add a reference [29] on line 38.

  1. Nakamura K, Toda T, Saruwatari H, et al. Speaking-aid systems using GMM-based voice conversion for electrolaryngeal speech[J]. Speech communication, 2012, 54(1): 134-146.

Point 5: lines 39 - 40 - citation of  Liu et al.  - there is not such author in the cited articles.

Response 5: Thank you for pointing out this. We have replaced references [7-8] and their description.

  1. Huang C, Tao X Y, Tao L, et al. Reconstruction of whisper in Chinese by modified MELP[C]//2012 7th International Conference on Computer Science & Education (ICCSE). IEEE, 2012: 349-353.
  2. Li J, McLoughlin I V, Song Y. Reconstruction of pitch for whisper-to-speech conversion of Chinese[C]//The 9th International Symposium on Chinese Spoken Language Processing. IEEE, 2014: 206-210.

Point 6: line 62 "The above methods are all rule-based whispered speech conversion techniques" but the citations use attention models and LSTM networks. So what is "rule based" according to authors?

Response 6: Thank you for pointing out this. We have replaced reference [12-13] and their description.

  1. Fan X, Lu J, Xu BL. Study on the conversion of Chinese whispered speech into normal speech [J]. Audio Engineering, 2005 (12): 44-47.
  2. Li XL, Ding H, Xu BL. Phonological segmentation of whispered speech based on the entropy function [J]. Acta Acustica, 2005 (1): 69-75.

Point 7: line 68 - citation 14-18 - are supposed to be about " GMM-based speech conversion methods to whispered speech conversion", but 15 and 16 are about special architecture of GMM (subspace GMM) and 18 is about speech enhancement.

Response 7: We agree with this suggestion and replace reference[15,16,18] as follows:

  1. Janke M, Wand M, Heistermann T, et al. Fundamental frequency generation for whisper-to-audible speech conversion[C]//2014 IEEE International Conference on Acoustics, Speech and Signal Processing (ICASSP). IEEE, 2014: 2579-2583. .
  2. Grimaldi M, Cummins F. Speaker identification using instantaneous frequencies[J]. IEEE transactions on audio, speech, and language processing, 2008, 16(6): 1097-1111.
  3. Li J, McLoughlin I V, Dai L R, et al. Whisper‐to‐speech conversion using restricted Boltzmann machine arrays[J]. Electronics Letters, 2014, 50(24): 1781-1782.

Point 8: Following citation is stated (liens 69 - 71)  : "Iwama et al. proposed to build a continuous probability model for the spectral envelope of whispered speech using a probability-weighted GMM [19] to obtain the mapping relations of channel parameters between the whispered speech and the corresponding normal speech"  The work does not not contain word whisper at all.

Response 8: Thank you for pointing out this. We have replaced the reference [19] as follows:

  1. Chen X, Yu Y, Zhao H. F0 prediction from linear predictive cepstral coefficient[C]//2014 Sixth International Conference on Wireless Communications and Signal Processing (WCSP). IEEE, 2014: 1-5.

Point 9: lines 100 - 102  "In the first stage, the reference normal speech and whispered speech were processed with dynamic time warping (DTW), and then the MFCC parameters of each frame of normal speech and whispered speech were extracted." what is used for DTW alignment, when MFCCs are extracted only after it?

Response 9: Thank you for pointing out this correction. It is corrected in the Section 2.1 as follows: "In the first stage, the MFCC parameters of each frame of normal speech and whispered speech were extracted, and then the reference normal speech and whispered speech were processed with dynamic time warping (DTW).”

Point 10: line 128 - upper indices needs to be mixed (xy, yx),line 151  - log operation is missing in the description MFCC computation,line 158-159 - "The speech frequency can be divided into a series of triangular filter sequences..." does not make any sense.

Response 10: Thanks for noticing these typos and grammatical mistakes. We have incorporated all the suggestions as follows:

line 128 - is corrected as

line 151 - log operation is added.

line 158-159 - Since the critical band changes with the frequency, the speech frequency is divided into a series of triangular filter sequences, the weighted sum of all signal amplitudes within the frequency bandwidth of each triangular filter is taken as the output of a band-pass filter, and then the output of all filters is further transformed by DCT to obtain MFCC.

Point 11: line 162: X is not explained

Response 11: We agree with this suggestion. It is corrected in the Section 2.2 as follows:

X is the frequency domain data of speech by STFT.

Point 12: line 169: "Because of the sparse characteristics of speech signals" -- what is meant by the "sparse characteristics"?

Response 12: It means that the coefficients corresponding to normal speech are often sparsely distributed.

Point 13: line 171: "Xu et al. [23]" -- there is no author Xu in article 23.

Response 13: Thank you for pointing out this mistake. We have exchanged reference [23] and reference [24].

Point 14: How many speakers were in the speech database?

Response 14: We have added a description of the dataset in the Section 3.1 as follows:

There were 10 speakers, each with 10 sentences, covering five subjects such as stock market, catering, tourism, sports and film.

Point 15: Did authors use different speakers in train and test set?

Response 15: We have added a description in the Section 3.1 as follows:

We randomly selected 90% whispered speech sentences and the corresponding parallel corpus as training data,the rest 10% was used as test data.

Point 16: How the pairs C, C' for Eq. 14 were created?

Response 16: We used the matlab’s Cepstral Feature Extractor block to extract cepstral coefficients from an audio file.

[audio,fs] = audioread('1.wav');

windowLength = round(0.04*fs);overlapLength = round(0.02*fs);S = abs(stft(audio,"Window",hann(windowLength,"periodic"),"OverlapLength",overlapLength,"FrequencyRange","onesided"));cc = cepstralCoefficients (designAuditoryFilterBank(fs,'FFTLength',windowLength)*S);

Point 17: The comparison with other methods is missing. It only compares different variant of the same technique.

Response 17: We agree with this suggestion and add comparison with DNN in the Section 3.2. We provided MOS with 95% confidence w.r.t LPC,DNN,MFCC and MFCCs.

Point 18: The proposed method novelty is only to use 3 consecutive frames for the processing, other parts of the method have already be published.

Response 18: A conversion method based on the inversion of Mel frequency cepstral coefficient (MFCC) features was proposed to convert whispered speech into normal speech. As shown in formula 11 we propose an double sparse constraint-based signal reconstruction model. In the experimental setup,we used an interframe overlap of 50%,3 frames of whispered speech and 1 frame of normal speech to constitute a joint feature vector.

Reviewer 2 Report

This paper proposes a method for converting whispered speech into normal speech. The method is based on a combined GMM that associates MFCC from whispered and normal speech. The proposed method does not have novelty because it is a well-known strategy in the research community. But the analyses carried out are interesting and the results are well presented. There are several aspects that I think can be improved before publication:

1.- In all the experiments, it is important to include more details about the experimental setup: amount of data, EM iterations, etc.

2.- It is not clear the total available data. I suggest including a table with all the figures. Also, it is important to know the language and the text pronounced in these recordings. The phoneme variability is one important aspect to evaluate the obtained results.

3.- I’d like to know how the data is split into training (GMM), validation and testing. Are you considering recordings from the same subject in training and testing? I think it is important to analyze the system in speaker independent scenarios: different speakers in training and testing.

4.- Could you provide statistically significance of the differences with results in figure 5?

5.- I am wondering if you have analyzed other strategies for transforming one MFCC distribution into another one. MLLR?

Author Response

Reviewer’s general remarks: This paper proposes a method for converting whispered speech into normal speech. The method is based on a combined GMM that associates MFCC from whispered and normal speech. The proposed method does not have novelty because it is a well-known strategy in the research community. But the analyses carried out are interesting and the results are well presented. There are several aspects that I think can be improved before publication.

Response: Thank you very much for the encouraging remarks on findings of the paper. The revised version has addressed all the comments and answered all the reviewer’s queries.

Point 1:  In all the experiments, it is important to include more details about the experimental setup: amount of data, EM iterations, etc.

Response 1: We agree with this suggestion and add a description in the Section 3.1 as follows:

There are 10 speakers (5 females and 5 males), each with 10 sentences, covering five subjects such as stock market, catering, tourism, sports and film. The length of each sentence was approximately 2 s, with a sampling rate of 8 kHz and a precision of 16 bit.

The number of EM iterations is 300.

Point 2: It is not clear the total available data. I suggest including a table with all the figures. Also, it is important to know the language and the text pronounced in these recordings. The phoneme variability is one important aspect to evaluate the obtained results.

Response 2: The language is chinese. There were 10 speakers, each with 10 sentences, covering five subjects such as stock market, catering, tourism, sports and film. Each utterance was sampled at 8 kHz, with 16-bit PCM storage,First, the frame segmentation was performed on the signals, with a frame length of 512 sampling points and an interframe overlap of 50%,3 frames of whispered speech and 1 frame of normal speech constitute a joint feature vector.

Point 3: I’d like to know how the data is split into training (GMM), validation and testing. Are you considering recordings from the same subject in training and testing? I think it is important to analyze the system in speaker independent scenarios: different speakers in training and testing.

Response 3: We randomly selected 90% whispered speech sentences and the corresponding parallel corpus as training data,the rest 10% was used as test data. The model does not consider speaker relevance.

Point 4: Could you provide statistically significance of the differences with results in figure 5?

Response 4: We agree with this suggestion and add a figure as follows:

MOS with 95% confidence w.r.t LPC,DNN,MFCC and MFCCs

Point 5: I am wondering if you have analyzed other strategies for transforming one MFCC distribution into another one. MLLR?

Response 5: The MLLR training strategy is not considered in this paper. We will study the impact of MLLR on the model in our future work.

Reviewer 3 Report

The whispered speech is a voiced sound with an energy approximately 20 dB lower than that of normal speech. The work propose an approach to convert whispered speech into normal speech by using a conversion method based on the inversion of Mel frequency cepstral coefficients (MFCC).

The paper improve former approaches and the tests seem to validate the advantages of this method, L1/2. 

Generally, the paper is well written and present a scientific soundness, however some improvements may be considered.

It is not obvious the statement at rows 92-93

Eqs. use different size fonts

Some details in the experimental setup may be omitted.

Fig.1 must be reviewed, legends of some figs may be improved and also the quality.

Instead of speeches may be utterances is suitable; matrices

Eq.14 need attention.

Please specify how this sapling frequency was chosen, row 207

Author Response

Reviewer’s general remarks: The whispered speech is a voiced sound with an energy approximately 20 dB lower than that of normal speech. The work propose an approach to convert whispered speech into normal speech by using a conversion method based on the inversion of Mel frequency cepstral coefficients (MFCC). The paper improve former approaches and the tests seem to validate the advantages of this method, L1/2. Generally, the paper is well written and present a scientific soundness, however some improvements may be considered.

Response: Thanks for these positive remarks on the paper. We have incorporated all the reviewer’s suggestions and answered all the queries.

Point 1: It is not obvious the statement at rows 92-93.

Response 1: When training GMM model, every 3 frames of whisper speech and 1 frame of normal speech constitute a joint feature vector, in which 3 frames of continuous whisper speech are used to consider the relationship between frames. It can be seen from formula 13 that the enery spectrum can be directly calculated from the MFCC spectrum without fundamental frequency estimation. It can be seen from Fig. 3 that the proposed method obtains the spectrograms similar to the reference normal speech without estimating the fundamental frequency. The harmonic component can be clearly seen from the spectrograms, indicating that the proposed method can obtain the spectrograms estimation of normal speech.

Point 2: Eqs. use different size fonts.

Response 2: We agree with this suggestion and check all Eqs.

Point 3: Some details in the experimental setup may be omitted.

Response 3: We agree with this suggestion and add a description in the Section 3.1 as follows:

There are 10 speakers (5 females and 5 males), each with 10 sentences, covering five subjects such as stock market, catering, tourism, sports and film. The length of each sentence was approximately 2 s, with a sampling rate of 8 kHz and a precision of 16 bit.

The number of EM iterations is 300..

Point 4: Fig.1 must be reviewed, legends of some figs may be improved and also the quality.

Response 4: We agree with this suggestion. Figure 1 is reviewed.

Point 5: Instead of speeches may be utterances is suitable; matrices.

Response 5: We often use word speech with speech signal processing. For example,TTS is the abbreviation of text to speech,ASR is the abbreviation of automatic speech recognition.

Point 6: Eq.14 need attention.

Response 6: We agree with this suggestion. It is corrected as follows:

Point 7: Please specify how this sapling frequency was chosen, row 207.

Response 7: 8kHz and 16KHz are commonly used, which can ensure the quality of reconstructed speech, but 8kHz can achieve almost the same results as 16KHz with less calculation.

Round 2

Reviewer 1 Report

line 40, 41 - the sentence ends with "to generate" - to generate what?

line 41 - GMM is statistical model, Although is belongs to machine learning. Better category would be "neural network based".

line 42 - Authors mention that there is a group of conversion techniques based on NN, but no citation. The NN based methods are left out from the intro completely. The fact that authors chose statistic method should not prevent them to do full introduction or at least give citation representing the work. Later in tab. 1 they compare to the DNN based conversion, but the work is nowhere mentioned.

line 183: equation 12 is broken

line 277 and around - there are two other techniques used in table as reference. It would be good to provide citation of the work as well as 2-3 sentences of explanation.

Since the authors experimental setup is quite small, and they are actually using the same speakers in both train and test, it would be good to check how the speakers are handled in the works authors are comparing to.

Authors should discuss or experimentally verify, how the system would behave on unseen speaker

Figure 5 is useless - one more line in tab. 1 for CD would be enough. Or is there any other message in the figure than the numbers? Why the same techniques have different colors in the figures a) and b)

Author Response

Response to Reviewer 1 Comments

Point 1: line 40, 41 - the sentence ends with "to generate" - to generate what?

Response 1: Thanks for noticing the grammatical mistake. It is corrected as follows:

“One is rule-based whisper conversion, mainly using empirical observation or statistical modeling to generate transformation rules.”

Point 2: line 41 - GMM is statistical model, Although is belongs to machine learning. Better category would be "neural network based".

Response 2: We agree with you. Neural networks work better than GMMs in most cases, but are more prone to overfitting than GMMs, and GMMs may work better when there is less data.

Point 3: line 42 - Authors mention that there is a group of conversion techniques based on NN, but no citation. The NN based methods are left out from the intro completely. The fact that authors chose statistic method should not prevent them to do full introduction or at least give citation representing the work. Later in tab. 1 they compare to the DNN based conversion, but the work is nowhere mentioned.

Response 3: We agree with this suggestion and add 1 reference [30] as follows:

  1. Lian H, Hu Y, Yu W, et al. Whisper to normal speech conversion using sequence-to-sequence mapping model with auditory attention[J]. IEEE Access, 2019, 7: 130495-130504.

Point 4: line 183: equation 12 is broken.

Response 4: Thank you for pointing out and the correction is carried out.

Point 5: line 277 and around - there are two other techniques used in table as reference. It would be good to provide citation of the work as well as 2-3 sentences of explanation.

Response 5: We agree with this suggestion and add 2 references [30-31] with their explanation as follows:

  1. Lian H, Hu Y, Yu W, et al. Whisper to normal speech conversion using sequence-to-sequence mapping model with auditory attention[J]. IEEE Access, 2019, 7: 130495-130504.
  2. Yang M, Qiu F, Mo F. A linear prediction algorithm in low bit rate speech coding improved by multi-band excitation model[J]. Acta Acustica, 2001, 26(4): 329-334.

LPC is used to extract partial correlation coefficients and build signal generation models for synthesis.

The network configurations of DNN-based were 60-120-60-120-60.

Point 6: Since the authors experimental setup is quite small, and they are actually using the same speakers in both train and test, it would be good to check how the speakers are handled in the works authors are comparing to.

Response 6: In different methods, we used the same test set and train set partition method.

Point 7: Authors should discuss or experimentally verify, how the system would behave on unseen speaker.

Response 7: We agree with this suggestion and add analysis in the experimental section as follows:

Also, under the condition that the pronunciation is standard chinese, if the training set and test set are different subjects, even unseen speaker, it has little impact on the final result.

Point 8: Figure 5 is useless - one more line in tab. 1 for CD would be enough. Or is there any other message in the figure than the numbers? Why the same techniques have different colors in the figures a) and b).

Response 8: We agree with this suggestion and redraw Figure 5 so that the same techniques have the same colors. The figure is as follows:

Reviewer 2 Report

I think the authors have done an important effort for improving the paper but I think there is one aspect that must be included. I'd like to ask a subject independent experiment: the subjects from training and testing sets are different. I think this analysis is important to see the robustness of the proposed method.

Author Response

Reviewer’s general remarks: I think the authors have done an important effort for improving the paper but I think there is one aspect that must be included. I'd like to ask a subject independent experiment: the subjects from training and testing sets are different. I think this analysis is important to see the robustness of the proposed method.

Response: Thanks for these positive remarks on the paper. We add analysis in the experimental section as follows:

Also, under the condition that the pronunciation is standard chinese, if the training set and test set are different subjects, even unseen speaker, it has little impact on the final result.

Round 3

Reviewer 2 Report

The authors have addressed properly my comments and the paper can be accepted.